# A Combined Virtual Electrode-Based ESA and CNN Method for MI-EEG Signal Feature Extraction and Classification

**DOI:** 10.3390/s23218893

**Published:** 2023-11-01

**Authors:** Xiangmin Lun, Yifei Zhang, Mengyang Zhu, Yongheng Lian, Yimin Hou

**Affiliations:** School of Automation Engineering, Northeast Electric Power University, Jilin 132012, China; xm_lun77@163.com (X.L.); 18604347075@163.com (Y.Z.); 16642406109@163.com (M.Z.); lianyongheng2022@163.com (Y.L.)

**Keywords:** brain–computer interface (BCI), electroencephalography (EEG), motor imagery (MI), EEG source analysis (ESA), convolutional neural network (CNN)

## Abstract

A Brain–Computer Interface (BCI) is a medium for communication between the human brain and computers, which does not rely on other human neural tissues, but only decodes Electroencephalography (EEG) signals and converts them into commands to control external devices. Motor Imagery (MI) is an important BCI paradigm that generates a spontaneous EEG signal without external stimulation by imagining limb movements to strengthen the brain’s compensatory function, and it has a promising future in the field of computer-aided diagnosis and rehabilitation technology for brain diseases. However, there are a series of technical difficulties in the research of motor imagery-based brain–computer interface (MI-BCI) systems, such as: large individual differences in subjects and poor performance of the cross-subject classification model; a low signal-to-noise ratio of EEG signals and poor classification accuracy; and the poor online performance of the MI-BCI system. To address the above problems, this paper proposed a combined virtual electrode-based EEG Source Analysis (ESA) and Convolutional Neural Network (CNN) method for MI-EEG signal feature extraction and classification. The outcomes reveal that the online MI-BCI system developed based on this method can improve the decoding ability of multi-task MI-EEG after training, it can learn generalized features from multiple subjects in cross-subject experiments and has some adaptability to the individual differences of new subjects, and it can decode the EEG intent online and realize the brain control function of the intelligent cart, which provides a new idea for the research of an online MI-BCI system.

## 1. Introduction

A Brain–Computer Interface (BCI) system collects electroencephalography (EEG) signals, retrieves the most representative human neurophysiological signals, classifies them through analysis, and then generates control commands for external devices. At the same time, external devices can also generate a corresponding signal to feed back to the human brain, thus realizing a “brain-computer interaction” [1]. It can exchange information directly between the brain and the outside world without relying on conventional methods such as human muscle tissue or peripheral nerves [2]. Currently, for patients with paralysis, spinal cord injury, epilepsy, or brain nerve damage, the realization of an interaction with the external environment is a subject to be solved in the field of medicine and control [3,4,5,6,7]. In particular, the BCI system of the motor cortex has become a hot topic for domestic and foreign scholars [8].

Motor Imagery (MI) has been shown to have similar brain activity to performing real movements. Without external stimulation, subjects can reinforce the brain’s compensatory function by imagining movements of the hands or feet [9,10,11,12,13,14,15]. The motor imagery-based brain–computer interface (MI-BCI) system is a technology that converts MI into controlling external devices or performing tasks by decoding activity signals in the subject’s brain associated with specific motor imagery [16,17,18]. The basic principle is to use the neural activity patterns of the cerebral cortex to decode the subject’s motion image [19,20,21]. When a subject imagines performing a certain movement, specific electrical signals are sent out in the motor execution area of the brain. These signals can be captured by neural sensors (such as an electroencephalogram, EEG), and after signal processing and analysis, they are finally converted into instructions that computers can understand [22,23,24].

### 1.1. Literature Survey

The Wadsworth research group in the United States pioneered MI research with the release of BCI2000, an open-source software platform that allows the design, implementation, evaluation, and verification of BCI systems, which has greatly contributed to the development of BCI technology [25,26]. The BCI research team of Graz University of Technology in Austria was the first to discover the ERD/ERS phenomenon in EEG and conduct BCI research to enable patients with cervical spinal cord injuries to control artificial neural prostheses and restore upper limb motor functions such as independent contact and grasping, and then, based on this, developed an MI-BCI system based on the left and right hands with a recognition accuracy of 82.5~90% [27,28].

In recent years, BCI groups have also carried out research on the identification of complex MI. In 2013, Garcia et al. combined MI and myoelectricity to control a robotic arm with six degrees of freedom [29]. In 2015, He et al. mapped the sensorimotor rhythm EEG backwards to the cortical source domain to improve the performance of the BCI system, allowing it to communicate with the outside world as an alternative to physiological neural pathways [30]. In 2015, Leeb et al. developed a BCI system to improve the independence of patients with dyskinesia by using a network shared control technique, and verified the feasibility of the system through the comparative performance experiment of nine patients and ten healthy individuals [31]. In 2019, Bartur G et al. studied the EEG of 14 patients with their first cerebral pawn, and found that the ERD amplitude of EEG was significantly correlated with the residual motor function of the upper limb in patients with mild paralysis [32]. Domestically, the group of Shangkai Gao and Xiaorong Gao from Tsinghua University is one of the earlier research teams of MI-BCI. They have achieved good results in the international BCI Competition many times, and have also developed several BCI systems. They designed a hybrid BCI spelling system that combines visual evoked potentials and MI potentials at the moment of motion onset. Subjects were trained to better master the manipulation of the character input system, and input accuracy could be as high as 93.3% [33].

In recent years, research on MI-EEG based on the network structure has also become a hot topic. Ma et al. proposed a novel temporal dependency learning convolutional neural network with an attention mechanism. The experimental results on the BCI Competition IV-2a (BCIC-IV-2a) and OpenBMI datasets show that the network outperforms the state-of-the-art algorithms and achieves the average accuracy of 79.48%, improved by 2.30% on the BCIC-IV-2a dataset. The network demonstrates that learning temporal dependencies effectively improves the MI EEG decoding performance [34]. Wang et al. proposed an MI-EEG classification method designed to improve the classification accuracy by combining Shannon complex wavelets and convolutional neural networks. BCI competition IV dataset 2b as a public motor imagination dataset was tested to prove the validation of this proposed method. The classification accuracy and kappa value were adopted to prove the superiority of the proposed method by comparing it with the state-of-the-art classification methods [35]. Through the review paper by Hamdi Altaheri et al., a lot of relevant information can also be learned, including discussions on three issues: Is preprocessing required for DL-based techniques? What input formulations are best for DL-based techniques? What are the current trends in DL-based techniques? And then, the current challenges and future directions are discussed [36].

MI-BCI technology has a wide range of applications in many fields. In rehabilitation medicine, MI-BCIs can be used to help restore motor function to patients with impaired function [37], such as stroke, spinal cord injury, or prosthetic subjects. By training them to use motor imagery to control external devices, patients can regain some degree of mobility and improve their quality of life [38]. In addition, MI-BCI can also be applied to fields such as virtual reality, human–computer interactions, and brain-controlled interfaces, providing subjects with a more natural and intuitive way of interacting [39,40]. Subjects can use imagination to control the actions of virtual characters, interact with computer interfaces, or participate in immersive gaming experiences. With the continuous advancement of technology, MI-BCI is expected to play an important role in medical rehabilitation, human–computer interactions, and other fields [41,42,43].

At present, many excellent MI-EEG feature extraction and classification theories and methods have emerged and achieved good results every year. However, from the practical application of MI-BCI, there are still many areas that need improvement. The main problems are as follows:

(i) Large individual differences. The MI-BCI system involves placing EEG electrodes on the cerebral scalp to obtain a signal. The differences in the brain structure and conductivity, electrode position errors, noise, and electromagnetic interference during the collection process often lead to significant differences between individual subjects. Even the same subject may have significant differences in their physiological and psychological states at different times. How to improve the generalization performance of the MI-EEG signal feature extraction and classification algorithm, enhance the adaptability and robustness to individual differences, and solve the problem of the poor performance of the cross-subject classification model has become one of the most challenging problems in current MI-BCI research.

(ii) Low classification accuracy. The MI-EEG signal is weak and has a low signal noise ratio (SNR). The collected original signal contains many noise-induced interference signals, that is, artifact signals, which are mainly noise interference generated by the external environment and other physiological activities of the human body. The low SNR of MI-EEG signals can lead to a low classification accuracy. In current studies, MI-EEG decoding paradigms are mostly limited to 2 to 4 classifications of hands and feet. For the two-classification problem of MI-EEG, many methods have achieved very good classification results, but for the multi-classification problem, the effect is not ideal. The more classification types, the lower the classification accuracy. How to improve the classification accuracy of multi-task MI-EEG remains to be studied further.

(iii) Poor online performance of the MI-BCI system. In addition to the low classification accuracy and large individual differences, the efficiency of the signal processing, data volume, and complexity of the algorithm are related to the poor online performance of the MI-BCI system. In order to achieve effective control, it is urgent to improve the decoding ability and fast response of MI-EEG. However, few existing algorithms can solve this problem.

An MI-EEG signal is a combined representation on the scalp surface of the electric potential and magnetic field generated by the electrophysiological activity of nerve cells inside the brain. It is usually measured using electrodes as sensors, which are fixed on the scalp surface and has a non-Euclidean spatial structure [44,45]. Currently, most MI-EEG classification algorithms are studied based on the data collected from various channels and identify and classify data according to one-dimensional or two-dimensional Euclidean structures, which may not fully reflect the spatial information of all MI-EEG channels [46]. To solve problems such as the large individual differences between subjects, low classification accuracy, and poor online performance of the MI-BCI system, it is crucial to fully utilize the spatial information of all MI-EEG channels and efficiently extract the motor intention from the MI-EEG signals [47].

In this paper, we take the spatial information of MI-EEG channels as a breakthrough and propose a combined virtual electrode-based ESA and CNN method for MI-EEG signal feature extraction and classification, which is used to explore the information about the electrophysiological activities of nerve cells inside the brain. The rest of the paper is organized as follows: Section 2 introduces data acquisition and preprocessing, ESA, feature extraction, and CNN classification. The results of the feature extraction, classification, and MI-BCI system experiments are presented in detail in Section 3. In Section 4, the impact of noise cancellation, training duration, and individual differences on the MI-BCI system was discussed, and the article was concluded with a summary.

### 1.2. Contributions

The main contributions are summarized as follows.

(i) The Boundary Elements Method (BEM) was used to solve the forward problem; the Low-Resolution Electromagnetic Tomography algorithm (LORETA) was used to constrain the inverse problem and construct the source model. The MI-EEG signals collected by electrodes were projected onto the cortex through EEG Source Analysis (ESA), and then virtual electrodes were constructed to obtain the source signals. Then, a Morlet wavelet was used for Joint Time Frequency Analysis (JTFA) to extract two-dimensional joint time frequency characteristics. In this way, the non-Euclidean spatial information of virtual electrodes can be converted into two-dimensional planar Euclidean information.

(ii) In the recognition and classification of the two-dimensional Euclidean structure data, CNN has the advantage of extracting multi-scale local features and combining them to improve the classification performance. In this paper, we proposed a combined virtual electrode-based ESA and CNN method for MI-EEG signal feature extraction and classification. The results indicate that the online MI-BCI system developed based on this method can improve the decoding ability of multi-task MI-EEG after training. At the same time, in interdisciplinary experiments, it can learn the generalized features of multiple subjects and has certain adaptability to the individual differences of new subjects, providing a new approach for the research of online MI-BCI systems.

## 2. Materials and Methods

### 2.1. Overview

The overall design framework of MI-BCI system is shown in Figure 1. The system mainly consists of EEG acquisition system, preprocessing, ESA, feature extraction, CNN classification, control strategy, wireless Bluetooth transmission, and a smart car system. Its working flow is as follows:(i)The EEG acquisition system was built using a portable electroencephalograph and supporting software, and the signals from 10 subjects performing 6 types of MI tasks (left fist, right fist, left foot, right foot, left thumb, and right thumb). Self-collected data set was established.(ii)A preprocessing module is used to remove external and internal biological noise interference in the MI-EEG signals.(iii)After preprocessing, the MI-EEG signals were mapped to the cortex through ESA, and 9 pairs of virtual electrodes were constructed to obtain the source signals. JTFA was performed to extract joint time–frequency feature information.(iv)A 6-classification CNN model was constructed, using a 4-layer CNN structure to learn signal features, 4-layer max-pooling for dimensionality reduction, and FC layer for classification of MI tasks.(v)The control strategy module converted the classification results of the CNN model into control instructions, and then transmitted them to the smart car via wireless Bluetooth.(vi)The motion state of the smart car is fed back to the subject for verification and judgment.(vii)Based on the self-collected data set, the experiments were conducted to verify the classification effect of the MI-BCI system, and the results were analyzed and optimized.

### 2.2. Data Acquisition and Preprocessing

The EEG acquisition system mainly consists of the Emotiv Epoc+ wireless portable electroencephalograph and supporting software, which adopts non-invasive EEG technology with built-in amplifier, filter, and analog-to-digital converter modules. It has 16 electrodes, placed according to the 10–20 international standard lead system, including 2 reference electrodes (P3, P4) and 14 measurement electrodes (AF3, F7, F3, FC5, T7, P7, O1, O2, P8, T8, FC6, F4, F8, and AF4). By default, the sampling frequency is set to 128 Hz and can be changed as needed. MI-EEG signal acquisition interface is shown in Figure 2, which can realize real-time display of EEG data stream. There are two data-receiving modes, wireless Bluetooth and USB, and the acquired MI-EEG signal can be converted into digital signal form and stored in the computer.

When preprocessing MI-EEG signals, using an 8–30 Hz bandpass filter can preserve information closely related to MI. This can remove artifacts caused by external noise and internal biological noise while extracting effective frequency band information. During signal acquisition, the inevitable blinking of the subject can cause interference from electro-oculographic artifacts, which are evident in the prefrontal region of the brain and can be removed using independent component analysis (ICA). Figure 3 shows 2D (2-dimensional) topographic map of the MI-EEG signals under ICA. The color change from blue to yellow to red in the figure is the process of energy transition from weak to strong. Due to the small number of electrodes in this device and the small number of fractions formed, the 2D brain topography has certain limitations, which can be further analyzed in combination with the 3D (3-dimensional) brain topography again, as shown in Figure 4. Combining the 2D and 3D brain topography, it can be determined that components **5** and **12** are higher in energy in the prefrontal region only, with significant electrical ophthalmic interference, and these components can be excluded from the MI-EEG.

### 2.3. EEG Source Analysis

Due to the accumulation of bioelectric signals generated by the interactions between neurons within the brain in the scalp layer, and the fact that source activity within the brain can only be obtained through derivation, the spatial characteristics of EEG signals are relatively limited [48]. The conduction relationship between the source activity inside the brain and EEG signals is extremely complex. The EEG source analysis (ESA) mainly includes the Forward problem and the Inverse problem. The Inverse problem is a process of deriving the source signal inside the brain using EEG signals collected on the scalp surface. To solve the Inverse problem, it is necessary to first solve the Forward problem, which is the process in which the source signals are attenuated by layers of cortex, skull, and scalp, and finally transmitted to the limited electrodes on the scalp surface. Solving the Forward problem relies on the creation of the head model and it also constrains the location of the source when solving the Inverse problem. In general, the relationship between source activity inside the brain and EEG signals collected from the scalp layer can be analyzed using a linear model [49].

#### 2.3.1. Forward Problem

The head model established in this paper is a 3-layer (scalp, skull, and cortex) head volume conduction model, which can better fit the real head model and improve the accuracy of source positioning [50]. The forward problem is to treat the brain as a quasi-static current field, and the source and head models are known to solve the scalp surface potential, essentially solving the transfer matrix [51]. Based on Maxwell, Poisson equation can be used for specific description, which is shown in Equation (1):(1)$$𝛻\cdot \left(\sigma 𝛻\phi \right)=-\sum_{\Omega }I_{s}$$
where σ denotes the electrical conductivity, $I_{s}$ is the volumetric current density generated by the neurons in the brain, Ω represents the entire field, and ϕ denotes the unknown electrical potential value developed on the scalp surface.

The boundary state of the scalp layer is shown in Equation (2):(2)σ(𝛻ϕ)⋅n=0

The boundary integral of the field is described using Green’s theorem and is shown in Equation (3):(3)$$2\pi \mu (p_{0})={∭}_{\Omega }\frac{\rho }{\sigma rp,p_{0}}dv-\iint_{\Gamma }\mu (p)\frac{𝜕}{𝜕np}(\frac{1}{rp,p_{0}})dsp+\iint_{\frac{𝜕}{𝜕np}}\frac{1}{rp,p_{0}}\frac{𝜕}{𝜕np}\mu (p)dsp$$
where $p_{0}$ is a point in a different dielectric partition interface, Ω is the field domain, Γ is the boundary of Ω, $rp,p_{0}$ is the distance between p and $p_{0}$, ρ is the density value of the source current per unit volume.

In this paper, BEM (boundary element method) is adopted to deal with the forward problem by discretizing the boundary integral [52]. The multi-dielectric field is divided into several regions. If there is only one dipole source in each region, the conductivity is $\sigma_{s}$, and the number of interfaces in this region is M, then the potential value of the point $p_{0}(r)$ on $S_{k}$ is shown in Equation (4):(4)$$u_{S_{k}}(r)=\frac{2\sigma_{s}}{\sigma_{k}^{-}+\sigma_{k}^{+}}u^{\infty }(r)+\frac{1}{2\pi }\sum_{l=1}^{M}\left(\frac{\sigma_{l}^{-}-\sigma_{l}^{+}}{\sigma_{k}^{-}+\sigma_{k}^{+}}\right)\iint_{S_{l}}u_{S_{l}}(r^{\prime })d\Omega^{kl}(r^{\prime })$$

Thus, $\sigma_{l}^{-}$ and $\sigma_{l}^{+}$ are the values of conductivity on each side of the interface $S_{l}$ and D is the current dipole of $r_{0}$.

The stereo angle of face $ndS_{l}$ opposite $S_{k}$ is shown in Equation (5):(5)$$d\Omega^{kl}\left(r^{\prime }\right)=\frac{r^{\prime }-r}{|r^{\prime }-r{|}^{3}}\cdot ndS_{l}$$

The potential of $p_{0}(r)$ is shown in Equation (6):(6)$$u^{\infty }\left(r^{\prime }\right)=\frac{D}{4\pi \sigma }\cdot \frac{r-r_{0}}{|r-r_{0}{|}^{3}}$$

The entire head volume of the brain is segmented into M sub-regions by a number of closed curved surfaces $S_{l}(l=1,2,\cdots ,M)$. To facilitate the solution of the integral equation, each interface $S_{k}$ of these closed surfaces is decomposed into smaller triangles $\Delta_{i}^{\left(k\right)}$ of quantity size $n_{k}$, which can be expressed in Equation (7):(7)$$S_{k}={⋃}_{i=1}^{n_{k}}{∆}_{i}^{\left(k\right)}\;\;\;\;\;\;\;\;\;\;\;\;\;\;\;k=1,\ldots ,M$$

The sum of the small triangles on all interfaces is shown in Equation (8):(8)$$N=k=\sum_{k=1}^{M}n_{k}$$

Let $n_{k}^{\prime }$ be the number of discrete nodes constructed by all small triangles on $S_{k}$ surface, then the total number of discrete points on the M interfaces is shown in Equation (9):(9)$$N^{\prime }=\sum_{k=1}^{M}n_{k}^{\prime }$$

Equation (4) becomes:(10)$$u_{S_{k}}\left(r\right)=\frac{2\sigma_{s}}{\sigma_{k}^{-}+\sigma_{k}^{+}}u^{\infty }\left(r\right)+\frac{1}{2\pi }\sum_{l=1}^{M}\frac{\sigma_{l}^{-}-\sigma_{l}^{+}}{\sigma_{k}^{-}+\sigma_{k}^{+}}\sum_{j=1}^{n_{l}}{\iint_{\Delta_{j}^{l}}u_{S_{l}}\left(r^{\prime }\right)d\Omega }^{kl}\left(r^{\prime }\right)$$

This step is equivalent to discretization of the integral equation, which can be further transformed into Equation (11):(11)U=G+BU
(12)$$U=\left[\begin{array}{c} u^{1}\\ \vdots \\ u^{M} \end{array}\right],\;G=\left[\begin{array}{c} g^{1}\\ \vdots \\ g^{M} \end{array}\right],\;B=\left[\begin{array}{c} B^{11}\;\;\;\cdots \;B^{1M}\\ \;\vdots \;\;\;\;\;\;\ddots \;\;\;\vdots \\ B^{M1}\;\cdots \;B^{MM} \end{array}\right]$$

Equation (11) is transformed into Equation (13):(13)$$u^{k}=g^{k}+\sum_{i=1}^{M}B^{kl}u^{l}(k=1,\cdots ,M)$$
where $u^{k}$ and $g^{k}$ are both column vectors of dimension $n_{k}$ and $u^{k}$ is the potential value at the center of gravity of each small triangle $\Delta_{i}^{\left(k\right)}$ on $S_{k}$.

#### 2.3.2. Inverse Problem

Contrary to the direct problem, the inverse problem is to infer and calculate the signal sources inside the brain, including the position, direction, intensity, etc., of the signal sources [53]. In order to solve the inverse problem, the distributed source model is used to locate the signal source in this paper. The entire cortical area of the brain in this model is divided into tightly connected and sufficiently small triangular blocks. Each small triangle contains electrical activity formed by dipoles.

Since the position of the triangle is fixed, it means that the spatial position of the dipole is fixed, and the distributed source model can be considered as a linear problem. The relationship between the three dipole distances of each dipole and the scalp potential values of each electrode is expressed in Equations (14)–(17):(14)u=KJ
(15)$$u={\left[u_{1},u_{2}\cdots u_{N}\right]}^{T}$$
(16)$$J={\left[j_{1},j_{2}\cdots j_{M}\right]}^{T}$$
(17)$$K=\left[\begin{array}{cccc} k_{11} & k_{12} & \cdots & k_{1M}\\ k_{21} & k_{22} & \cdots & k_{2M}\\ \cdots & \cdots & \cdots & \cdots \\ k_{N1} & k_{N2} & \cdots & k_{NM} \end{array}\right]$$

In Equation (15), u is the potential on the scalp electrode, and N is the number of electrodes.

In Equation (16), J is the current density of the signal source in the brain, 3M is the dimensional column vector, M is the number of signal sources, and the spatial position of signal sources is shown in Equations (18) and (19):(18)$$r=\left(x,y,z\right)$$
(19)$$J_{m}=[j_{mx},j_{my},j_{mz}{]}^{T},\;\left(m=1,\cdots M\right)$$

In Equation (17), K represents the conduction matrix of order N×3M, and the effect of the m-th dipole on the n-th electrode on the scalp surface is shown in Equation (20):(20)$$k_{nm}=\left[j_{nmx},j_{nmy},j_{nmz}\right],\;\left(n=1,\cdots ,N;m=1,\cdots ,M\right)$$

The contribution of the dipole inside the brain to the potential generated by the n-th electrode on the scalp surface is shown in Equation (21):(21)$$u_{n}=\sum_{m=1}^{M}u_{nm}=\sum_{m=1}^{M}k_{nm}j_{m}=\sum_{m=1}^{M}\left(k_{nmx}j_{mx}+k_{nmy}j_{my}+k_{nmz}j_{mz}\right)$$
where $u_{nm}$ is the potential generated by the m-th dipole on the n-th electrode, and it can be obtained as shown in Equation (22):(22)$$k_{nmx}=\frac{u_{nm}}{j_{mx}},k_{nmy}=\frac{u_{nm}}{j_{my}},k_{nmz}=\frac{u_{nm}}{j_{mz}}$$

In this model, a large number of fixed dipoles are used to represent the distribution of electrical activity in the brain. The number of dipoles is larger than the number of electrodes, and the calculation process produces infinite solutions, that is, highly underdetermined.

The underdetermined problem of source computation can be solved by adding some constraints. The common methods include Minimum Norm Estimate (MNE), Weighted Minimum Norm Estimate (WMNE), LORETA algorithm, etc. In this paper, the LORETA algorithm is chosen to constrain the inverse problem to solve the underdetermined problem.

According to the characteristics of strong synchronous electrical activity of adjacent neurons, this algorithm introduces Laplacian operator of the graph, which can not only significantly promote spatial smoothness constraint, but also obtain three-dimensional spatial distribution information of brain dipoles [54]. To solve the inverse problem under constraints, u=KJ is shown in Equation (23):(23)$$\min{\left\|J\right\|}^{2}=\min J^{T}CJ$$
where, for a symmetric positive definite square matrix $C=C_{J}^{-1}$, the weighted minimum L2-norm solution is shown in Equation (24):(24)$$J_{WMNE}=C^{-1}K^{T}{\left(KC^{-1}K^{T}\right)}^{-1}u$$

When the coordinates of the reference electrode are not known, the constraint is shown in Equation (25):(25)$$u=KJ+cl_{N}$$
where $l_{N}$ is the N-dimensional column vector, the elements are all 1, and C is any unknown constant.

The solution can be solved by first fixing c, finding J, and then minimizing c. That is, Equation (25) is equivalent to Equation (26):(26)Hu=HKJ
where $H=I-\frac{1}{N}l_{N}l_{N}^{T}$, and the solution of the formula is shown in Equation (27):(27)$$J_{WMNE}=C^{-1}G^{T}{\left(GC^{-1}G^{T}\right)}^{+}u$$
where, G=HK is a diagonal square matrix of order 3M×3M, $C=C_{J}^{-1}={\left(WW^{T}\right)}^{-1}$, and the complementing bias matrix W is shown in Equation (28):(28)$$W=diag(\left\|k_{1}\right\|,\left\|k_{2}\right\|,\cdot \cdot \cdot ,\left\|k_{3M}\right\|)$$

Combining the Laplace operator matrix L of the graph with the complementary bias matrix W as a weighting matrix, by substituting Equation (27), Equation (29) is obtained:(29)$$J_{LORETA}={\left(WL^{T}LW\right)}^{-1}G^{T}{\left(G{\left(WL^{T}LW\right)}^{-1}G^{T}\right)}^{+}u$$

In order to improve the stability and anti-interference of the estimated results, Tikhono regularization is performed on Equation (21) and Equation (26) to obtain Equation (30):(30)$$J_{LORETA}=\underset{\lambda }{\arg\min}(||u-KJ-cl_{N}|{|}^{2}+\lambda ||LWJ|{|}^{2})$$

The solution of Equation (29) is shown in Equation (31):(31)$$J_{LORETA}={\left(WL^{T}LW\right)}^{-1}G^{T}{\left(G{\left(WL^{T}LW\right)}^{-1}G^{T}+\lambda H\right)}^{+}u$$
where λ is the given adaptive parameter.

Compared to other methods, LORETA algorithm performs depth localization estimation by reducing the expected resolution, coupled with smooth constraints, which not only accurately locates surface sources in 3-dimensional solution space, but also greatly improves the accuracy of deep source localization [55].

### 2.4. Feature Extraction

Electrodes of non-Euclidean spatial structure record the combination of electrical signals transmitted to the scalp from a source composed of numerous clusters of neurons inside the brain [50]. To precisely decode the source activity inside the brain, MI-EEG signals needs to be traced back into the brain via ESA. The specific steps can be described as follows:

Firstly, the data of the head model are imported, and the three-layer (scalp, skull, and cortex) head volume conduction model is constructed to determine the direction, position, and intensity of the dipole inside the brain. In this paper, the direction of the dipole is set perpendicular to the surface of the cortex. Then, BEM algorithm is used to solve the forward problem, the mathematical model of the brain electric field is established, and the expression for the scalp potential of the dipole in the distributed source model is constructed. The parameters of the dipole are adjusted to fit the scalp potential calculated by the forward problem to the MI-EEG data. Based on the fitting results, a source model is generated and the LORETA algorithm is used to constrain the EEG inverse problem. Figure 5 shows the schematic diagram of head model and source model. Finally, 18 virtual electrodes closely related to the MI task are identified in the cortex region of the source model. The range of each virtual electrode is extended to 20 vertices, each with a constrained dipole, which is considered as the source. Figure 6 shows the position of the selected virtual electrodes and the extracted time waveform information from the two virtual electrodes.

The virtual electrodes of the left and right hemispheres are LP1–LP9 and RP1–RP9, respectively, forming 9 virtual electrode pairs P1–P9. JTFA is performed on the MI signal of the virtual electrode to extract the joint time–frequency feature information of 5 s. The transverse coordinate of 2D data is time, and the transverse dimension is 640 at the sampling frequency of 128 Hz. The longitudinal coordinate is frequency, band of interest of 8–30 Hz extracted by band-pass filter, and the longitudinal dimension is 23. Therefore, the MI time–frequency feature size of a virtual electrode is 640 × 23, and then the data of the symmetric virtual electrodes of the left and right hemispheres are stitched longitudinally with a size of 640 × 46. The time–frequency map size of 640 × 46 × 9 for 9 pairs of virtual electrodes is used as the input to the CNN model.

### 2.5. CNN Classification

CNN uses a series of filtering, normalization processes, and nonlinear activation functions to extract features from large training datasets, which is more suitable for complex MI-EEG signal processing [56,57,58,59]. The basic components of CNN include convolution layer, pooling layer, and fully connected layer.

The function of the convolution layer is to extract signal features, and the convolution operation mainly adopts Sparse Connectivity and Shared Weights, which can greatly reduce the number of CNN parameters and accelerate the training speed [60,61,62]. Pooling layer can reduce the data dimension and the complexity of network computing, accelerating the computing speed [63,64]. After feature extraction, FC layer is used to perceive global information and complete classification. Adaptive Moment Estimation (Adam) is adopted as the optimizer to minimize the loss function. In addition, weight and bias are updated through the back propagation algorithm, and the learning rate of the optimizer is 1 × 10^−5^ [65].

In this paper, a deep CNN classification model is constructed based on the self-collected 6-classification MI-EEG datasets, and the parameters are shown in Table 1. A 4-layer CNN is used to learn MI-EEG feature information, the activation function selects Leaky ReLU function, the 4-layer Pool uses max pooling to reduce dimensions, then the data are flattened by Flatten layer to get 1D data, and finally the FC layer uses the softmax function to achieve classification prediction. Each layer is convolved with Batch Normalization (BN) and 50% dropout to reduce the risk of network overfitting, and Adam with learning rate 1 × 10^−5^ is used as the optimizer to minimize the loss value.

## 3. Results

The training of the MI-BCI system is an important link, which usually requires subjects to participate in the feedback and adjustment process. During the process of training, the subject is asked to imagine a specific motor image, and the system continuously optimizes the model by comparing the recorded EEG signal with the subject’s intention to improve the accuracy and efficiency of decoding [66,67,68].

In this study, 10 subjects aged between 21 and 28 were selected, including seven males and three females. Subjects need to be trained to make the performance of the MI-BCI system meet the classification requirements. The training environment of the BCI system is shown in Figure 7, where one computer is used to display the stimulus task, while the other computer runs EmotivPRO to collect MI-EEG signals.

The training period of the system was 7 days, and each subject trained for 5 cycles per day, performing 6 MI tasks per cycle and 10 experiments per task. The 6 MI tasks are: left fist (T1), right fist (T2), left foot (T3), right foot (T4), left thumb (T5), and right thumb (T6).

The timing diagram of single MI training is shown in Figure 8, and the training time was 8 s. At the 0th second, the system generated a prompt tone, followed by pictures of the left fist, right fist, left foot, right foot, left thumb, and right thumb on the screen, guiding the subject to imagine the corresponding body movements according to the pictures: “left fist clenched”, “right fist clenched”, “extend left foot”, “extend right foot”, “erect left thumb”, and “erect right thumb”; at the 5th second, the pictures disappeared and the subject relaxed; at the 8th second, this training ended and the subject was prompted to move on to the next MI task.

### 3.1. Denoising Results

An MI-EEG signal is a very weak bioelectrical signal that is susceptible to external interference (such as power frequency interference, electrostatic interference, and electromagnetic interference). It will also mix with other physiological signals of the subject, such as electromyography and electrocardiogram signals, and even contain many components with unknown physical meanings. In this study, the raw MI-EEG signals collected by the Emotiv Epoc+ wireless portable electroencephalograph also contained interference from external and subject-specific factors. Before the signal feature extraction and classification, 8–30 Hz band-pass filtering and 2D and 3D brain topographic map analysis are required for the raw MI-EEG signals.

Figure 9 is a set of MI-EEG signal waveforms of subject S1 during task T2 (right fist). Figure 9a is the raw MI-EEG signal waveform, and Figure 9b is the preprocessed MI-EEG signal waveform. Where the vertical axis represents 14 collecting electrodes and the horizontal axis represents time. The comparison shows that the preprocessed MI-EEG signal waveform image is clearer and more concise, and the interference of high-frequency noise is reduced, which improves the signal-to-noise ratio to a certain extent.

### 3.2. Feature Extraction Results

After the ESA calculation of 14 channels of MI-EEG, nine pairs of virtual electrodes were constructed on the cortex to obtain the source signals. A Morlet wavelet was performed to extract the joint time–frequency feature information. A set of time–frequency feature maps of subject S1 in task T2 (right fist) is shown in Figure 10, including the time–frequency data of nine pairs of symmetrical virtual electrodes in the left and right brain. The horizontal axis of each time–frequency map is the time range of 0~5 s, and the vertical axis is the frequency range of 8~30 Hz. The graph shows the intensity of the MI-EEG signal at different times and frequencies, with the color tending to blue representing lower energy and the color tending to red representing higher energy. The two areas of the cortex most associated with MI (somatomotor and somatosensory areas) are controlled and sensed by the contralateral torso, i.e., the left side of the brain corresponds to the right half of the torso, while the right side of the brain corresponds to the left half of the torso. The comparison of the left and right brain time–frequency maps shows that the energy of the left brain is stronger than that of the right brain when subject S1 performs the right fist MI.

### 3.3. Classification Results

In this paper, 10 groups of single-subject experiments were conducted, and 900 samples were collected from each subject and divided into five equal parts. The training set (540 samples), validation set (180 samples), and test set (180 samples) were divided in the ratio of 3:1:1. All samples in the training and validation sets were sent to the CNN network through uniform normalization. Afterwards, the model was trained based on the training set and the optimal parameters were selected on the validation set to achieve stable convergence and obtain the optimal model. Then the test set was fed into the optimal model to obtain the test results, and the average of the five results was used as the classification result. This can reduce the randomness caused by data partitioning and help improve the stability of the model.

Figure 11 shows the loss curves of 10 individual subjects obtained by the CNN model during training. Their loss values decrease continuously with the increase of the number of iterations, and reach the minimum value and remain basically stable at about 800 iterations, so as to obtain the optimal training effect. It can be seen from the figure that the CNN model trained on the data of 10 individual subjects is convergent.

In this paper, the classification performance of the BCI system on the self-collected data set of individual subjects is measured by the common evaluation metrics of accuracy, precision, recall, and the F1-score, as shown in Table 2. As can be seen from the table, subject S1 has the best test performance and S8 has the worst. The mean values of accuracy, precision, recall, and the F1-score for the 10 subjects are 81.08%, 82.77%, 92.07%, and 87.13%, respectively. The results verified that the BCI system achieved a good classification effect on the self-collected data sets of 10 subjects. Subjects with classification accuracy, from high to low, were S1, S7, S10, S3, S6, S4, S9, S2, S5, and S8, and the top six subjects all achieved more than 80% accuracy, and they were selected to participate in the next session of the testing experiment.

Figure 12 shows the accuracy histograms of six MI tasks for 10 subjects, with the highest accuracy of 90.75% (S1) and the lowest of 71.25% (S8) for T1; the highest accuracy of 93.23% (S1) and the lowest of 71.58% (S8) for T2; the highest accuracy of 87.60% (S7) and the lowest of 71.14% (S8) for T3; the accuracy ranged from a high of 92.91% (S1) to a low of 71.45% (S8) for T4; T5 had a high of 83.29% (S1) and a low of 66.85% (S8); and T6 had a high of 84.70% (S1) and a low of 66.92% (S8). The average accuracy of the six MI tasks was 84.41% (T1), 85.59% (T2), 82.67% (T3), 83.75% (T4), 74.01% (T5), and 76.05% (T6), ranked as T2 (right fist) > T1 (left fist) > T4 (right foot) > T3 (left foot) > T6 (right thumb) > T5 (left thumb). The best MI task for classification is the right fist, and the worst is the left thumb.

Figure 13 shows the ROC curve and AUC value of a single subject obtained when the CNN model was tested, from which it can be seen that the average AUC value for the 10 subjects was 0.945; S1 had the largest AUC value of 0.963 and S8 had the smallest AUC value of 0.920. Then, the CNN model achieved a good classification performance on the self-collected dataset of 10 subjects, S1 had the best classification effect after training, and S8 had the worst effect.

In summary, the six-classification CNN model constructed and trained by the method proposed in this paper achieved high classification accuracy on the self-collected datasets of 10 subjects, indicating that the method has a good generalization performance and can effectively complete the task of EEG intention classification.

### 3.4. MI-BCI System Experiment Result

After system debugging, the hardware and software of the MI-BCI are connected properly and the communication is in good condition. The control strategy module will continuously accumulate the classification results output by the CNN model, and convert the results into corresponding control instructions only when they reach the set threshold, which will be transmitted wirelessly to the smart car through the Bluetooth module. Table 3 shows the corresponding control functions and instructions of the six MI tasks in the BCI system.

In this paper, the intelligent car developed by our research group is selected as the external controlled device, which is mainly composed of a Microcontroller Unit (MCU) module, a drive motor control module, a steering gear control module, a speed detection module, a path information acquisition module, a power module, and a wireless Bluetooth transmission module.

The online test experiment of the MI-BCI system is carried out based on the trained optimal CNN model. During the experiment, the smart car was placed on the designed track, which was a square of 1 m in length and width surrounded by black tape on the white background floor. The car was required to run the whole course at a uniform speed according to the subjects’ brain intention. The online test environment of the MI-BCI system was shown in Figure 14.

Before the BCI system experiment, the steering gear angle and speed and threshold parameters of the intelligent car need to be set. The track is square with four 90-degree turns and the rest is straight. In order to turn the car smoothly and reduce the impact of the car turning left and right in a straight line by misoperation, the maximum angle of the steering gear is set at 23 degrees. Since it takes 5s for the BCI system to collect the MI-EEG signal, the car speed and threshold should not be set too high. Choose two values for the speed, 2 cm/s and 4 cm/s, and three values for the threshold, 1, 2, and 3. Six modes are then determined:

Mode 1: Speed is 2 cm/s and threshold is 1;

Mode 2: Speed is 2 cm/s and threshold is 2;

Mode 3: Speed is 2 cm/s and threshold is 3;

Mode 4: Speed is 4 cm/s and threshold is 1;

Mode 5: Speed is 4 cm/s and threshold is 2;

Mode 6: Speed is 4 cm/s and threshold is 3.

Subjects S1, S3, S4, S6, S7, and S10 were selected to participate in the BCI system experiment. Each subject carried out six modes of training experiments, respectively, and in each mode, the car completed five laps clockwise and five laps counterclockwise.

Table 4 shows the statistics of the number of laps completed by the car under the control of six subjects in different modes. The number of complete clockwise laps of the car was 22, 26, 21, 25, 10 and 3, totaling 107, while the corresponding counterclockwise laps were 18, 20, 18, 20, 8 and 1, totaling 85. In different modes, six subjects performed better in clockwise than counterclockwise direction, which also verified the experimental results of MI-EEG signal classification, where T2 (right fist) was classified more accurately than T1 (left fist), and so the right-turn command was more accurate than the left-turn command when translated into car commands. The total number of laps in each mode is as follows: 40 laps in mode 1, 46 laps in mode 2, 39 laps in mode 3, 45 laps in mode 4, 18 laps in mode 5, and 4 laps in mode 6. It can be seen that mode 2 and mode 4 were completed well, with the total number of laps being 46 and 45, respectively. These two parameter settings can be selected for the online testing of the BCI system.

After the training, the relevant parameters were selected, and the adaptive coordination between the subjects and the intelligent car was completed. The online test experiment of the BCI system can be carried out. The experiment consisted of six subjects running three sets of tests in two selected modes, each consisting of a clockwise and counterclockwise lap.

Table 5 shows the test results in mode 2. The mean time and the mean total number of instructions for subject S1 to control the car to complete a set of tests were 561.8 s and 58, respectively, 655.0 s and 64 for subject S3, 696.7 s and 69 for subject S4, 685.1 s and 66 for subject S6, 672.8 s and 65 for subject S7, and 708.8 s and 70 for subject S10. In this mode, it can be seen that the average of the total number of instructions is proportional to the average of the time spent, with subject S1 taking the shortest time, having the least total number of instructions, and the best test result.

Table 6 shows the test results in mode 4. The mean time and mean total number of instructions for subject S1 to control the car to complete a set of tests were 484.0 s and 68, respectively, 544.1 s and 80 for subject S3, 581.8 s and 82 for subject S4, 525.4 s and 78 for subject S6, 520.2 s and 75 for subject S7, and 569.0 s and 81 for subject S10. It can be seen that the test effect of subject S1 is also the best in mode 4.

The speed of Table 5 is 2 cm/s and the speed of Table 6 is 4 cm/s. To complete the same test, the time used in the former should be theoretically twice as long as the time used in the latter. However, from the data listed in the two tables, it can be seen that the time used in Table 5 is obviously less than the two times of Table 6. In addition, the number of executed instructions listed in Table 5 is less than that in Table 6, because the threshold value of Table 6 is 1, and the car will execute the instruction immediately after receiving it once. While the threshold value of Table 5 is 2, the car will execute the instruction only when it receives the same instruction twice in a row, so the response will be slow, but it can better eliminate some wrong operations.

Figure 15 shows the car motion track recorded by subject S1 when he controlled the car to run the whole course under two modes. The orange dots in the figure is the starting position of the car and the blue dots in the figure is the motion track of car. It can be seen that, in the same mode, the track curve of the clockwise direction has less fluctuation than that of the counterclockwise direction, and it fits the track of black line more closely. This is because T2 (right fist) has higher classification accuracy than T1 (left fist), so the corresponding right turn instruction is more accurate than the left command, and the motion track of the clockwise fist is better than the counterclockwise fist. In mode 2, although the setting speed of the car is relatively low and the time used is longer, the threshold is set to two, which filters out some misoperations. The motion track of the car in mode 2 is better and has less deviation than that in mode 4.

To sum up, the parameters have different effects on the system. When setting a high threshold value, the flexibility of the car will also be reduced although it can reduce some misoperation. When setting a high speed, although the overall running time will be reduced, the deviation caused by the wrong instruction is greater and the effect of the motion trajectory is not good. The test effect cannot be compared directly with the different parameters set and the parameters should be selected according to the actual situation. When the parameters were the same, T2 (right fist) had a higher classification accuracy than T1 (left fist), and the right turn instruction was more accurate than the left turn instruction. The time used, the total number of instructions, and the track curve all verified that the subject performed better in the clockwise direction than the counterclockwise direction. In general, the experiment time is proportional to the total number of corresponding instructions. The fewer instructions received by the subjects, the shorter the time spent, and the better the test effect.

## 4. Discuss

When MI-EEG acquisition and data transmission are delayed, the signal processing rate and other factors are fixed, and many factors such as the EEG denoising effect, individual differences of the subjects, and the training effect of the subjects will affect the classification performance of the BCI system.

Based on the raw MI-EEG data and the preprocessed denoising data, 10 groups of comparison experiments were conducted in this study. The denoising effect of MI-EEG signals was measured by the evaluation indicators of accuracy, precision, recall, and the F1-score. The incremental values of each performance indicator after denoising are shown in Table 7. From the contents of the table, the maximum increment of 20.51% (S7) and the minimum increment of 9.96% (S2) was for accuracy, the maximum increment of 19.60% (S6) and the minimum increment of 10.44% (S8) was for precision, the maximum increment of 26.46% (S7) and the minimum increment of 17.28% (S1) was for recall, and the F1-score had a maximum increment of 22.60% (S7) and a minimum increment of 16.02% (S1). The mean incremental values of accuracy, precision, recall, and the F1-score for the 10 subjects were 15.02%, 15.55%, 21.94%, and 18.54%, respectively. According to the experimental results, the performance indicators of the proposed BCI system on a single subject dataset are significantly improved after filtering and other preprocessing of the original MI-EEG signals. The preprocessing operation adopted in this paper can effectively eliminate the interference of external factors and the subjects’ own factors on MI-EEG signals and improve the signal-to-noise ratio.

### 4.1. Training Duration Effect

MI-EEG is a kind of spontaneous signal that does not require external stimulation and reinforces brain compensatory functions by imagining limb movements without involving actual muscle movements. After a period of training, subjects can significantly activate their brain regions. In this paper, 10 subjects were selected to participate in the BCI system training for 7 days, and subjects trained for five cycles per day, performing six MI tasks per cycle, and 10 experiments for each task. The experimental data of subject S1 with the best training effect and subject S8 with the worst training effect were selected to analyze the influence of the factor of the training duration on the classification performance of the BCI system.

Figure 16 presents the daily training results for subjects S1 and S8. Subject S1 was trained daily, and accuracy was improved by 7.96%, 5.91%, 2.69%, 1.78%, −0.41%, and 0.46%; precision was improved by 9.75%, 7.44%, 1.37%, 0.38%, 0.32%, and 0.05%; recall was improved by 6.47%, 6.81% 4.08%, 2.75%, 0.60%, and −0.06%; and the F1-score was improved by 8.18%, 7.13%, 2.69%, 1.50%, 0.46%, and −0.01%. Subject S8 was trained daily, and accuracy was improved by 3.18%, 2.19%, 0.74%, −0.07%, 0.35%, and −0.08%; precision was improved by 3.41%, 1.89%, 2.16%, 0.93%, −0.18%, and 0.02%; recall was improved by −0.71%, 4.32%, 6.73%, 3.90%, 0.70%, and −0.20%; and the F1-score was improved by 1.63%, 2.97%, 4.11%, 2.14%, 0.17%, and −0.07%. After 7 days of training, the indicators for S1 of accuracy, precision, recall, and the F1-score were improved by a total of 18.39%, 19.31%, 20.65%, and 19.95%. S8 improved by a total of 6.31%, 8.23%, 14.74%, and 10.95%. From the experimental results, it can be seen that the performance of each indicator of the subjects after training was significantly improved, and only the improvement effect was different. For subjects in the first 5 days of training, the effect was obvious and the curve of each indicator tended to level off in the last 2 days, which indicates that the training had reached the optimal state.

### 4.2. Individual Difference Effect

In this study, three sets of cross-subject experiments were conducted to analyze the influence of individual differences on the classification performance of the BCI system. In the experiment, the dataset (9000 samples) of 10 subjects (S1–S10) was mixed first, and the CNN network training was carried out to obtain the optimal model. Then, three new subjects (S11, S12, and S13), including two males and one female, aged between 21 and 28 years old, were selected and 900 samples were collected for each subject. Finally, the CNN optimal model was tested for its classification performance to evaluate the adaptability of the BCI system to the individual differences of the new subjects.

Table 8 lists the classification performance evaluation indicators of the BCI system for cross-subject experiments, and it can be seen that subject S13 had the best performance with an accuracy of 82.11%, subject S12 had the worst performance with an accuracy of 74.11%, and the average value of accuracy reached 78.07%. The mean values of the other indicators of precision, recall, and the F1-score also reached 77.87%, 78.64%, and 78.23%.

Figure 17 shows the confusion matrix of the classification accuracy of six MI tasks for the three subjects, and it can be seen that the highest classification accuracy was T2 (85.67%) for S13 and the lowest accuracy was T5 (71.87%) for S12. The average accuracy of the six MI tasks were (Hou Y) 79.33% (T1), 82.06% (T2), 77.81% (T3), 78.56% (T4), 74.96% (T5), and 75.70% (T6), respectively. The average classification errors for T1 were 4.94% (T2), 4.20% (T3), 4.35% (T4), 3.48% (T5), and 3.70% (T6), respectively. The average classification errors for T2 were 4.96% (T1), 3.67% (T3), 3.77% (T4), 2.74% (T5), and 2.80% (T6), respectively. The average classification errors for T3 were 3.69% (T1), 4.09% (T2), 5.25% (T4), 4.36% (T5), and 4.80% (T6), respectively. The average classification errors for T4 were 4.03% (T1), 3.28% (T2), 5.30% (T3), 4.06% (T5), and 4.77% (T6), respectively. The average classification errors for T5 were 4.44% (T1), 4.13% (T2), 3.69% (T3), 4.55% (T4), and 8.23% (T6), respectively. The average classification errors for T6 were 3.55% (T1), 1.49% (T2), 5.33% (T3), 3.53% (T4), and 10.40% (T5), respectively.

The experimental results showed that the BCI system achieved good average accuracy and a low classification error on the six MI tasks, with the highest classification accuracy for the right fist (T2) and the lowest for the left thumb (T5). According to the confusion matrix, the kappa values of the three subjects were 0.732 (S11), 0.699 (S12), and 0.785 (S13), respectively, with an average value of 0.737. This indicates that the classification algorithm of the BCI system has good consistency in the new subject dataset and can effectively eliminate the effect of random classification.

In summary, the six-class CNN model of the BCI system learned generalized features from the dataset of 10 subjects, achieving a good classification and generalization performance on the dataset collected by new subjects, adapting to individual differences in subjects, and breaking through the limitations of the poor compatibility of other BCI systems with different subjects.

### 4.3. Comparison with Other Works

At present, the MI-BCI system has become a hot research topic for scholars both domestically and internationally, with new research results emerging every year.

The public PhysioNet 109-subject dataset was selected for experiments, and a total of 10 sets of experiments were conducted. The average value was taken as the classification result. The classification accuracy of the method proposed was compared with the other literature research results, as shown in Table 9.

Handiru V S et al. proposed an Iterative Multiobjective Optimization for the Channel Selection (IMOCS) algorithm. On the PhysioNet dataset, the SVM classifier achieved an average classification accuracy of 63.62% for the two MI tasks [69]. Youngjoo K et al. proposed the strong uncorrelating transform complex common spatial patterns (SUTCCSP) algorithm. The performance of multiple classifiers was evaluated based on PhysioNet’s two-MI dataset, in which the random forest (RF) classifier achieved the best classification accuracy of 80.05% [70]. Ma X et al. proposed a method based on Recurrent Neural Networks (RNNs) that can perform the parallel decoding of spatial and temporal information. An average classification accuracy of 68.20% was achieved on PhysioNet’s four-MI dataset [71]. In an end-to-end DL model constructed by Hauke Dose et al., the CNN model classifies the original MI-EEG signals without special feature extraction. On PhysioNet’s two-, 3-, and 4-MI datasets, the mean accuracy reached 86.49%, 79.25%, and 68.51%, respectively [13]. Hou Y et al. used a CNN model for classification based on Brain-source Imaging (ESI), and achieved an accuracy of 94.5% on the four-MI dataset of PhysioNet [72]. A hybrid optimization technique of the Flower Pollination algorithm and β-hill-climbing algorithm was proposed by Alyasseri Z et al. An accuracy of 96.05% was obtained using the SVM classifier [73].

In this paper, a combined virtual electrode-based ESA and CNN method is proposed, with an average classification accuracy of 97.83%, which is better than the classification performance of the other literatures in the Table 9, indicating that this method is effective in motion intention classification and can improve the decoding ability of MI-EEG signals.

## 5. Conclusions

In this paper, an MI-EEG acquisition system based on the Emotiv Epoc+ EEG electroencephalograph and supporting software was built, a combined virtual electrode-based ESA and CNN method was proposed for MI-EEG signal feature extraction and classification, and then an online MI-BCI system was constructed. Based on a six-classification self-collection dataset, an optimal CNN classification model with the required accuracy was trained and the MI-BCI system was tested online. Through the analysis of the experimental results such as the classification performance, the EEG denoising effect, the training duration, and individual differences, it was verified that the MI-BCI system has achieved a good online application performance after training. At the same time, it can achieve the control function of the brain over intelligent cars, providing technical reference for similar research topics.

## Figures and Tables

**Figure 1 sensors-23-08893-f001:**
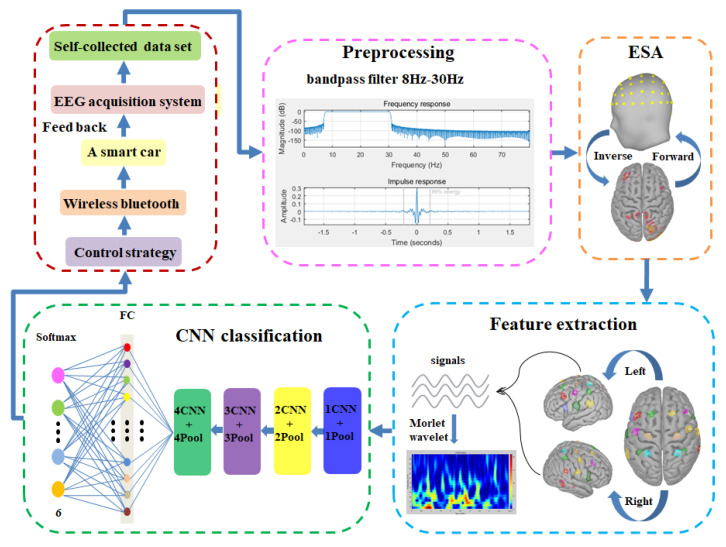
The framework of MI-BCI system.

**Figure 2 sensors-23-08893-f002:**
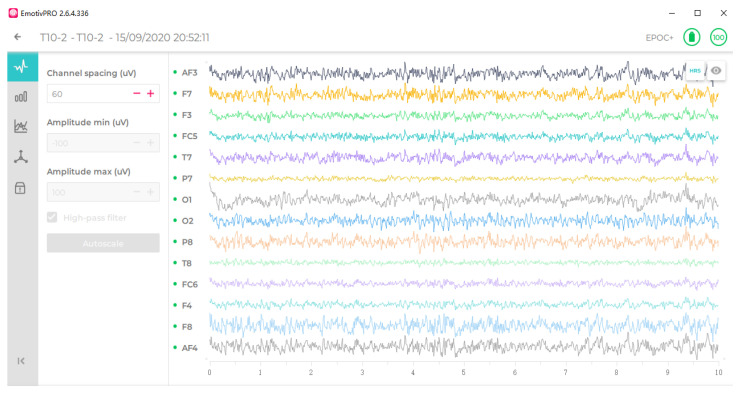
MI-EEG signal acquisition interface.

**Figure 3 sensors-23-08893-f003:**
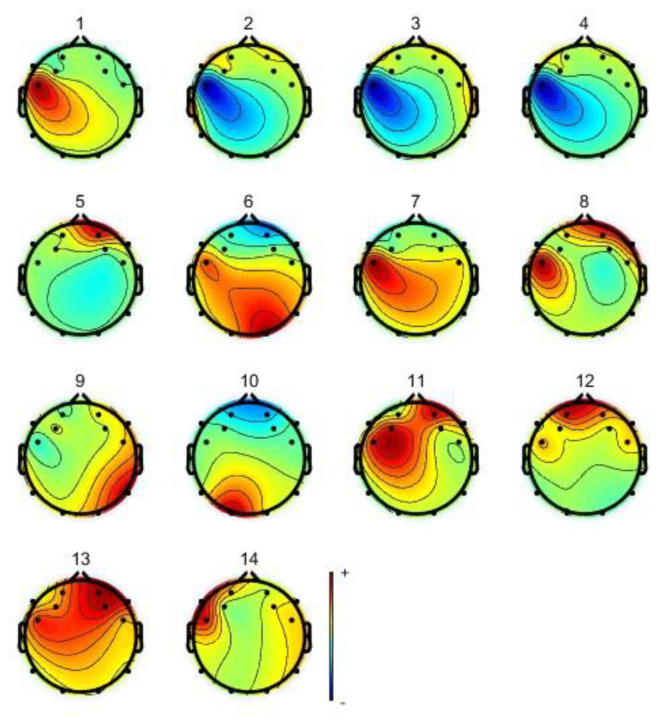
2D brain topographic map under ICA.

**Figure 4 sensors-23-08893-f004:**
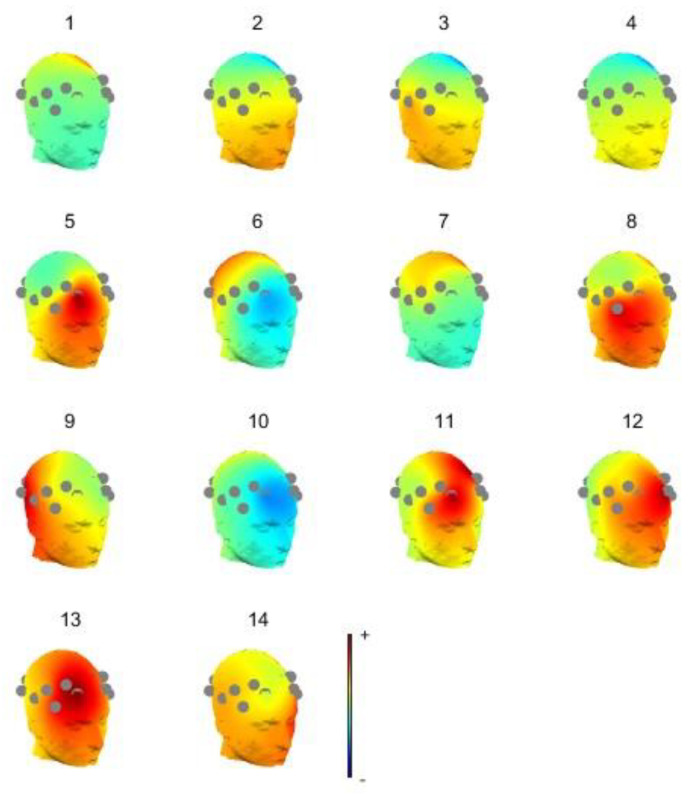
3D brain topographic map under ICA.

**Figure 5 sensors-23-08893-f005:**
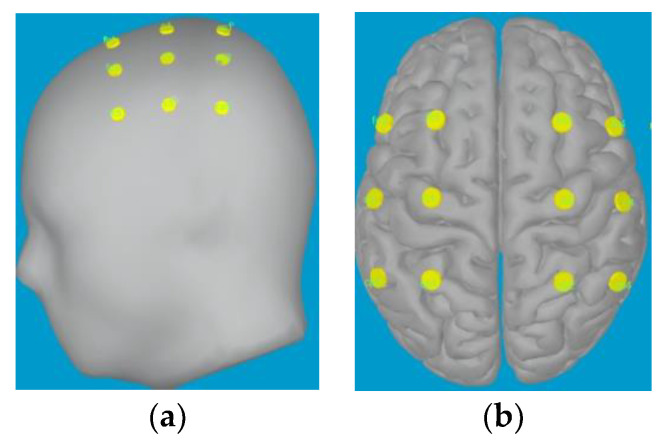
Schematic diagram of head model and source model. (**a**) Head model (**b**) Source model.

**Figure 6 sensors-23-08893-f006:**
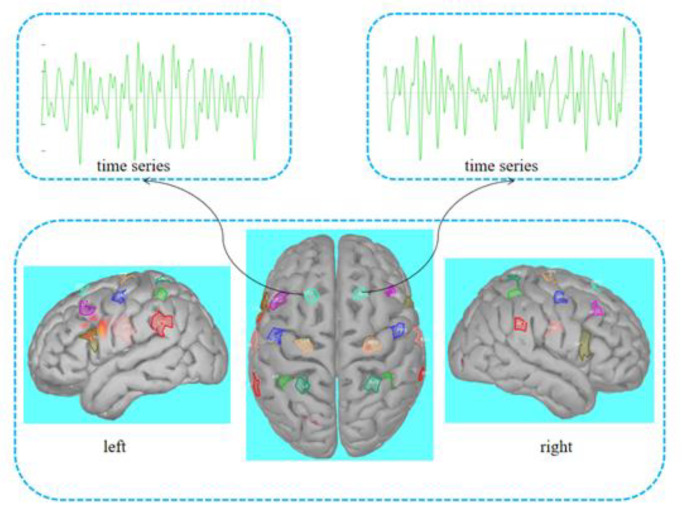
Selection of virtual electrodes and time waveform extraction.

**Figure 7 sensors-23-08893-f007:**
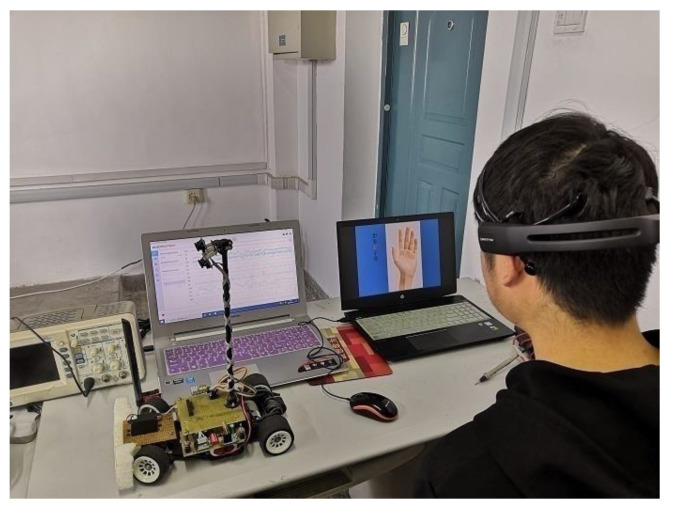
Training environment of BCI system.

**Figure 8 sensors-23-08893-f008:**
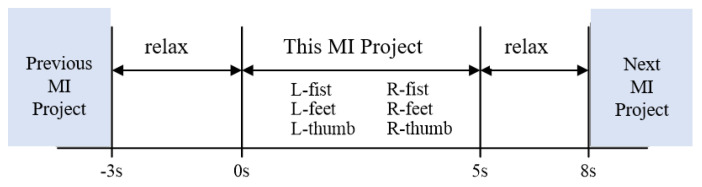
Timing diagram of single MI training for BCI system.

**Figure 9 sensors-23-08893-f009:**
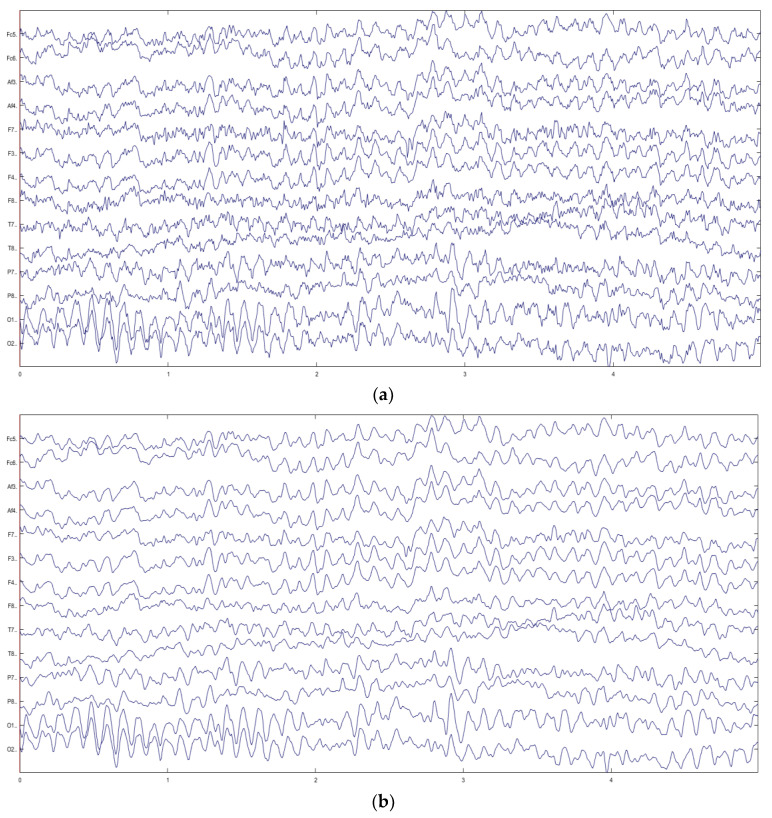
A set of MI-EEG signal waveforms of subject S1 during task T2 (right fist). (**a**) Raw MI-EEG signal waveforms. (**b**) Preprocessed MI-EEG signal waveforms.

**Figure 10 sensors-23-08893-f010:**
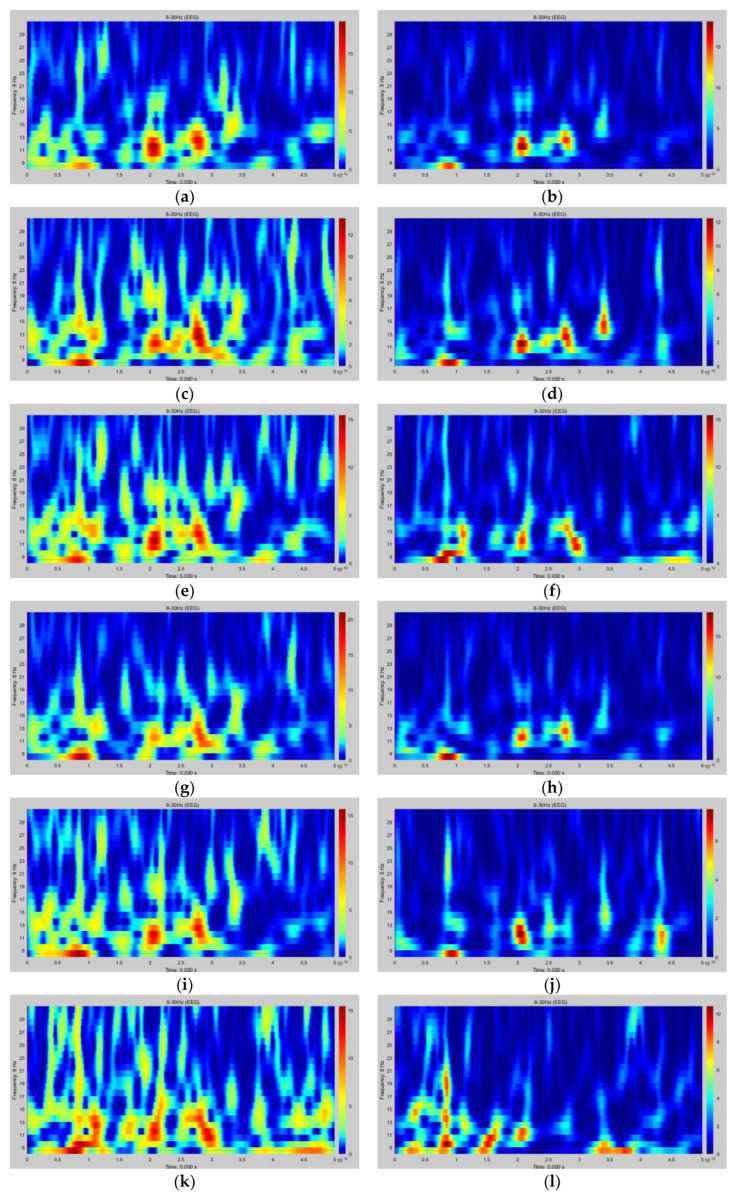
A set of time–frequency feature maps of subject S1 in task T2 (right fist). (**a**) S1T2#LP1; (**b**) S1T2#RP1; (**c**) S1T2#LP2; (**d**) S1T2#RP2; (**e**) S1T2#LP3; (**f**) S1T2#RP3; (**g**) S1T2#LP4; (**h**) S1T2#RP4; (**i**) S1T2#LP5; (**j**) S1T2#RP5; (**k**) S1T2#LP6; (**l**) S1T2#RP6; (**m**) S1T2#LP7; (**n**) S1T2#RP7; (**o**) S1T2#LP8; (**p**) S1T2#RP8; (**q**) S1T2#LP9; and (**r**) S1T2#RP9.

**Figure 11 sensors-23-08893-f011:**
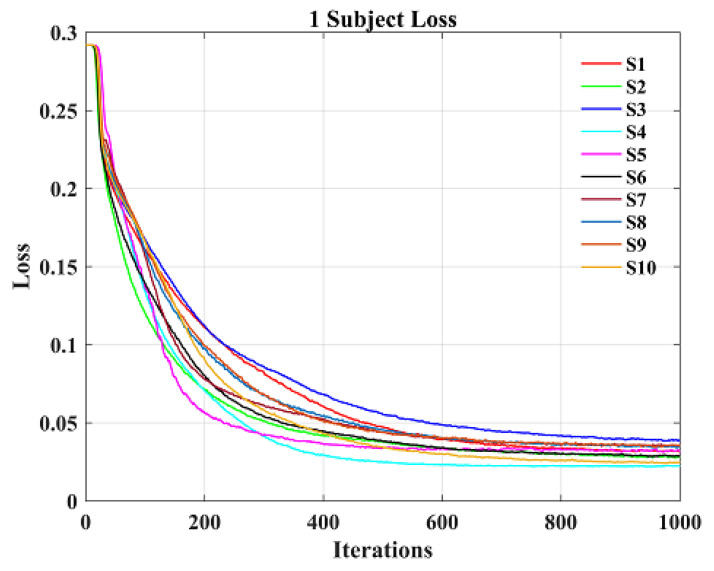
Loss curve of single subject.

**Figure 12 sensors-23-08893-f012:**
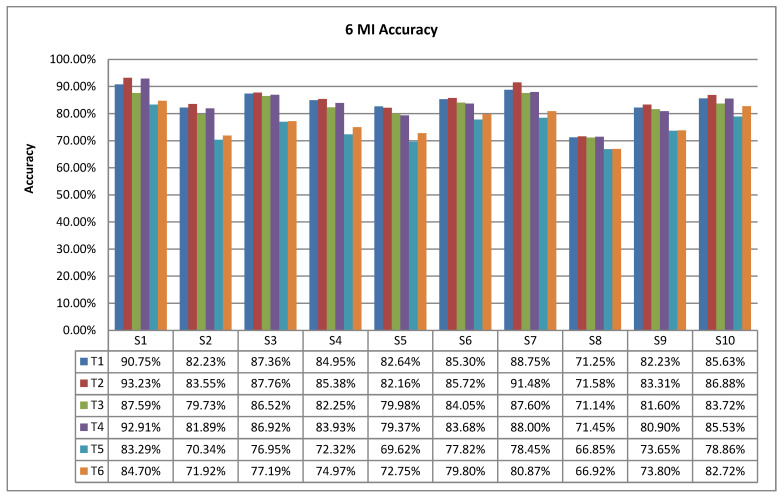
Accuracy histogram of 6 MI tasks.

**Figure 13 sensors-23-08893-f013:**
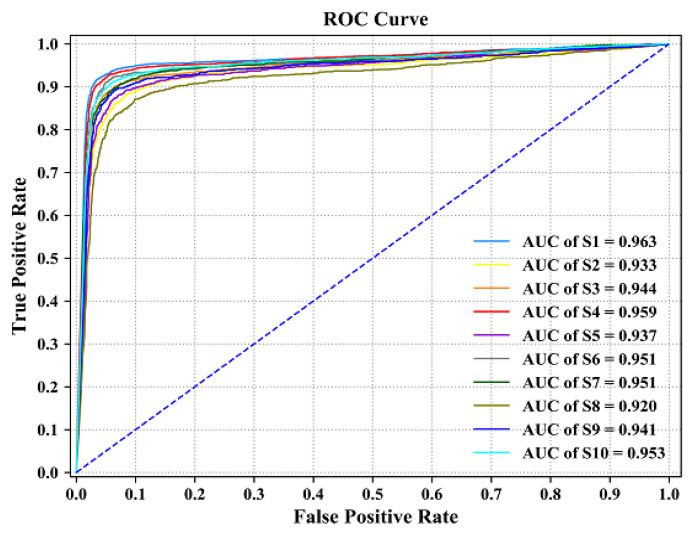
ROC curve and AUC value of single subject.

**Figure 14 sensors-23-08893-f014:**
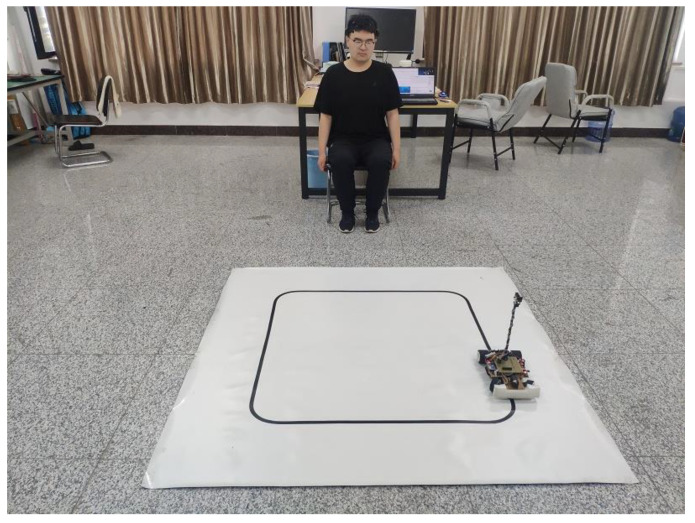
The online test environment of MI-BCI system.

**Figure 15 sensors-23-08893-f015:**
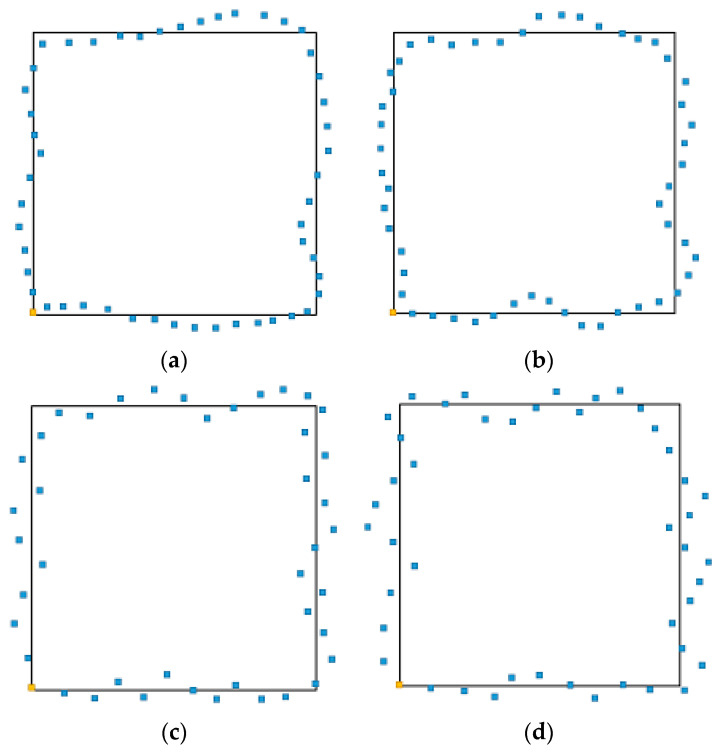
The car motion track. (**a**) Mode 2 clockwise; (**b**) Mode 2 counterclockwise; (**c**) Mode 4 clockwise; and (**d**) Mode 4 counterclockwise.

**Figure 16 sensors-23-08893-f016:**
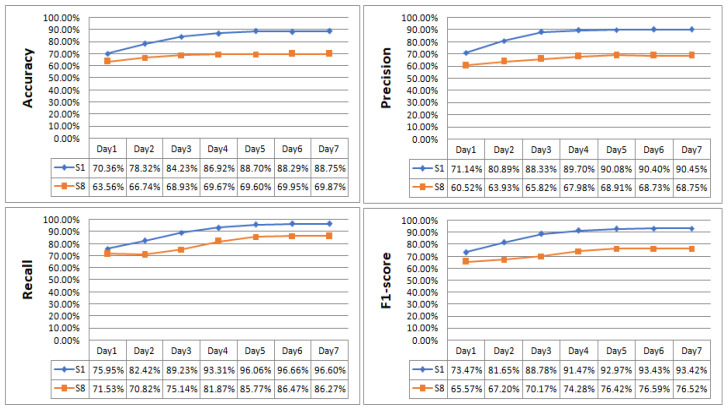
Daily training results for subjects S1 and S8.

**Figure 17 sensors-23-08893-f017:**
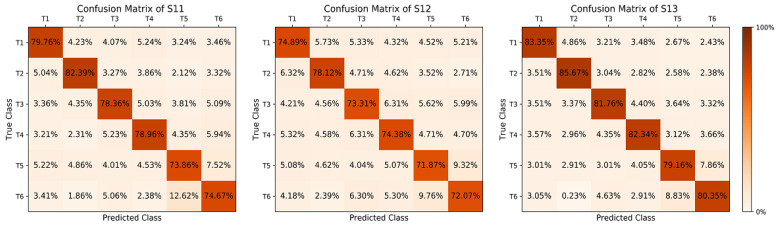
Confusion matrix.

**Table 1 sensors-23-08893-t001:** 6-Classification CNN model parameters.

Layer	Input Size	Map	Convolution Kernel Size	Pooling Size	Activation Function	Stride	Padding	Output Size
Input	640 × 46 × 9	1	-	-	-	-	-	640 × 414
CNN1	640 × 414	25	[1, 46]	-	Leaky ReLu	[1, 46]	Valid	640 × 9 × 25
Pool1	640 × 9 × 25	-	-	[3, 1]	-	[3, 1]	Valid	213 × 9 × 25
CNN2	213 × 9 × 25	25	[1, 9]	-	Leaky ReLu	[1, 1]	Valid	213 × 25
Pool2	213 × 25	-	-	[3, 1]	-	[3, 1]	Valid	71 × 25
CNN3	71 × 25	50	[11, 25]	-	Leaky ReLu	[1, 1]	Valid	61 × 50
Pool3	61 × 50	-	-	[3, 1]	-	[3, 1]	Valid	20 × 50
CNN4	20 × 50	100	[11, 50]	-	Leaky ReLu	[1, 1]	Valid	10 × 100
Pool4	10 × 100	-	-	[3, 1]	-	[3, 1]	Valid	3 × 100
Flatten	3 × 100	1	-	-	-	-	-	300
FC	300	1	-	-	-	-	-	6
Output	6	1	-	-	-	-	-	-

**Table 2 sensors-23-08893-t002:** Classification performance evaluation metrics for individual subjects.

Subject	Accuracy	Precision	Recall	F1-Score
S1	88.75%	90.45%	96.60%	93.42%
S2	78.28%	78.57%	92.44%	84.94%
S3	83.78%	86.39%	93.38%	89.75%
S4	80.63%	83.57%	90.70%	86.99%
S5	77.75%	78.99%	90.83%	84.50%
S6	82.73%	84.83%	93.18%	88.81%
S7	85.86%	87.42%	94.96%	91.03%
S8	69.87%	68.75%	86.27%	76.52%
S9	79.25%	82.48%	89.68%	85.93%
S10	83.39%	86.30%	92.65%	89.36%
Average	81.08%	82.77%	92.07%	87.13%

**Table 3 sensors-23-08893-t003:** Intelligent car control functions corresponding to MI tasks.

MI	Control Function	Control Instruction
R-thumb	start	0 × 01
L- thumb	stop	0 × 02
R-feet	forward	0 × 04
L-feet	backward	0 × 08
R-fist	right	0 × 10
L-fist	left	0 × 20

**Table 4 sensors-23-08893-t004:** Statistics of the number of laps completed by the car under the control of 6 subjects in different modes.

Subject	Mode 1		Mode 2		Mode 3		Mode 4		Mode 5		Mode 6	
S1	4	3	5	4	4	3	4	4	2	2	0	0
S3	3	3	4	4	3	3	5	4	2	1	1	0
S4	4	4	4	3	4	3	3	2	1	2	0	0
S6	4	3	5	3	3	2	4	3	2	1	1	0
S7	4	2	4	2	4	4	5	3	1	0	1	0
S10	3	3	4	4	3	3	4	4	2	2	0	1
SUM	22	18	26	20	21	18	25	20	10	8	3	1

**Table 5 sensors-23-08893-t005:** Test results in mode 2.

Subject	Group	Time (s)	Total Number of Commands	Start	Stop	Forward	Backward	Left	Right
	1	616.7	63	4	2	19	5	18	15
S1	2	522.5	55	3	2	24	4	12	10
	3	546.3	58	2	2	25	5	12	12
	1	670.9	65	3	3	18	6	19	16
S3	2	706.1	70	3	4	17	6	23	17
	3	588.0	59	3	2	22	5	15	12
	1	696.2	65	3	3	15	6	19	19
S4	2	759.8	75	2	3	10	7	29	24
	3	634.2	69	2	2	13	5	22	25
	1	635.7	61	3	3	17	7	15	16
S6	2	674.2	67	4	3	15	5	20	20
	3	745.5	72	4	2	15	7	21	23
	1	752.0	70	3	3	11	7	24	22
S7	2	700.5	67	2	2	11	6	21	25
	3	566.0	59	2	3	21	4	15	14
	1	643.0	63	2	3	16	7	16	19
S10	2	803.9	79	4	4	11	7	28	25
	3	679.6	68	2	3	15	6	23	19

**Table 6 sensors-23-08893-t006:** Test results in mode 4.

Subject	Group	Time (s)	Total Number of Commands	Start	Stop	Forward	Backward	Left	Right
	1	535.1	72	7	7	17	6	17	18
S1	2	473.3	68	6	7	16	5	18	16
	3	443.5	65	5	5	20	4	17	14
	1	526.8	77	6	5	15	7	24	20
S3	2	586.9	85	8	9	12	7	23	26
	3	518.7	79	8	8	15	5	23	20
	1	632.5	93	11	9	11	5	31	26
S4	2	578.2	82	9	9	15	8	22	19
	3	534.7	73	6	4	15	7	21	20
	1	549.2	87	9	7	13	7	25	26
S6	2	528.0	75	7	8	16	8	19	17
	3	499.1	72	5	5	17	4	22	19
	1	503.8	78	9	8	15	7	20	19
S7	2	510.1	72	7	8	18	5	18	16
	3	546.8	75	9	8	15	5	20	18
	1	612.0	90	12	13	13	7	26	19
S10	2	542.2	79	9	8	18	7	18	19
	3	552.9	75	7	7	17	5	22	17

**Table 7 sensors-23-08893-t007:** The incremental values of each performance indicator after denoising.

Subject	Accuracy	Precision	Recall	F1-Score
S1	+16.54%	+14.87%	+17.28%	+16.02%
S2	+9.96%	+12.90%	+21.72%	+16.84%
S3	+18.46%	+17.81%	+21.94%	+19.77%
S4	+10.33%	+13.36%	+22.44%	+17.77%
S5	+17.14%	+16.78%	+19.35%	+17.97%
S6	+13.41%	+19.60%	+19.28%	+19.51%
S7	+20.51%	+19.06%	+26.46%	+22.60%
S8	+13.66%	+10.44%	+25.06%	+16.80%
S9	+11.91%	+17.16%	+21.08%	+19.01%
S10	+18.30%	+13.55%	+24.77%	+19.13%

**Table 8 sensors-23-08893-t008:** The classification performance evaluation indicator of the BCI system for cross-subject experiment.

Subject	Accuracy	Precision	Recall	F1-Score
S11	78.00%	77.36%	78.75%	78.05%
S12	74.11%	72.73%	76.79%	74.70%
S13	82.11%	83.52%	80.40%	81.93%
Average	78.07%	77.87%	78.64%	78.23%

**Table 9 sensors-23-08893-t009:** Comparison of classification accuracy with the other literature.

Literature	MI Task	Average Accuracy	Dataset	Method
Handiru V S et al. [69]	2	63.62%	PhysioNet	IMOS + SVM
Youngjoo K et al. [70]	2	80.05%	PhysioNet	SUTCCSP + RF
Ma X et al. [71]	4	68.20%	PhysioNet	RNNs
	2	86.49%		
Hauke Dose et al. [13]	3	79.25%	PhysioNet	CNN
	4	68.51%		
Hou Y et al. [72]	4	94.50%	PhysioNet	CNN
Alyasseri Z et al. [73]	4	96.08%	PhysioNet	SVM
This work	4	97.83%	PhysioNet	ESA + CNN

## Data Availability

Publicly available datasets were analyzed in this study. This data can be found at: https://github.com/KJHGVS/MI-EEG-dataset.git (accessed on 29 October 2023).

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
