# Peer review of "A Combined Virtual Electrode-Based ESA and CNN Method for MI-EEG Signal Feature Extraction and Classification"

_sensors, 2023, doi:10.3390/s23218893_

Round 1

Reviewer 1 Report

Comments and Suggestions for Authors

The research titled "A combined virtual electrode-based ESA and CNN method for MI-EEG signal feature extraction and classification" appears to focus on the application of two different approaches for processing EEG  signals related to MI. In this research, two techniques, Virtual Electrode (ESA), and CNN, have been combined for feature extraction and classification of EEG signals.  To enhance the quality and applicability of your MI-BCI system, a significant effort is recommended in optimizing the EEG signal. This involves noise reduction and increasing the signal-to-noise ratio. Simultaneously, consider the possibility of expanding the sample size to address individual differences and improve result generalization. This will not only enhance the model's reliability but also boost classification accuracy, which is crucial for the success of the brain-computer interface in clinical and rehabilitation applications.

Comments on the Quality of English Language

No comments

Author Response

Thank you for the valuable suggestion, we have made changes according to the suggestions.

The preprocessing of MI-EEG signals is described in Section 2.2, where an 8-30Hz band-pass filter is used to retain information closely related to MI, while extracting effective frequency band information and removing artifacts caused by external noise and internal biological noise. During signal acquisition, the inevitable blinking of the subject can cause interference from electrooculographic artifacts, which can be removed using independent component analysis (ICA). In this paper, 8-30Hz band-pass filtering processing of raw MI-EEG signals as well as 2D and 3D brain topography analysis are used. Added 3.1 denoising results in the results section, introducing the waveform of the MI-EEG signal before and after denoising preprocessing. Through comparison, it can be seen that the preprocessed MI-EEG signal waveform image is clearer and more concise, and the interference of high-frequency noise is reduced, which improves the signal-to-noise ratio to a certain extent. In section 4.1, 10 groups of comparison experiments are conducted to measure the denoising effect of MI-EEG signal by the evaluation indicators of accuracy, precision, recall and F1-score. From the experimental results, it can be seen that the BCI system proposed in this paper significantly improves all the performance indicators on the single-subject dataset. It shows that the preprocessing operation adopted in this paper can effectively eliminate the interference of external and subject's own factors on the MI-EEG signal and improve the signal-to-noise ratio.

According to the suggestion to expand the sample size, this paper has added 4.4 Comparison with other works. The public PhysioNet 109-subject dataset was selected for experiments in this section, and a total of 10 sets of experiments were conducted. The average value was taken as the classification result. The average classification accuracy reached 97.83%, which is better than the research results of other literature listed in Table 9. The results indicate that the method proposed in this paper is effective in motion intention classification and can improve the decoding ability of MI-EEG signals.

For a revised manuscript,please see the attachment.

Reviewer 2 Report

Comments and Suggestions for Authors

This paper presents a CNN-based method for classification of MI-EEG. While the topic of the paper is interesting I have some major concerns regarding the paper which should be addressed before being further considered for publishing:

1. Although the paper provides quite an extensive list of references, most references are older than 10 years, and there are almost no references after 2020. The authors should report on the more recent work in the area of MI-EEG, several examples (but not limited to) being:

https://www.sciencedirect.com/science/article/abs/pii/S1568494622007347

https://ieeexplore.ieee.org/abstract/document/10196350

Authors can also consult this overview paper: https://link.springer.com/article/10.1007/s00521-021-06352-5

2. Authors have a paper from 2020 titled "A Simplified CNN Classification Method for MI-EEG via the Electrode Pairs Signals" which deals with the same topic of MI-EEG and use of CNNs for classification. However, in this paper authors do not explain the difference (novelty) from the previous paper which is required to determine the level and significance of contributions from this paper.

3. The dataset collected in this paper seems a bit limited: only 10 subjects, all young adults, with great majority of men. Why did the authors create their own dataset and only used that one (any specific reason)? Why wasn't any data used from Physionet? Is this self-collected dataset publicly available?

4. It is unclear what tasks named "left fist", "right fist", etc. actually encompass. What is required of the subject in order to complete for example "left fist" task?

5.Why use a smart car? It is unclear how it is related to the tasks like "left fist", etc. Are they used to control the motion of the car?

6.I'm a bit confused with the section 4.1. which discusses the denoising effect. Why aren't these results considered before in the results section. Do the authors consider the denoising as a regular part of their MI-EEG analysis or is it done only in specific cases. This needs to be clarified. 

7. Considering the results overall, there is a lack of comparison with other similar work. The reported level of accuracy (and other measures) suggests there is a lot of room for improvement. The authors should discuss that and also compare with the state of art.

Additional minor improvements:

- There is too much text devoted to explaining basic concepts like Maxwell's laws, neural networks, etc. This should be removed and keep the focus of the paper on describing the novel contributions and methods employed by this paper.

- References are missing DOI identifiers

Comments on the Quality of English Language

The paper requires an extensive English editing. There are numerous grammar mistakes, and some sentences are quite difficult to understand.

Author Response

Thank you for the valuable suggestion, we have made changes according to the suggestions.

1.Thank you very much for the following publications. We have carefully studied these articles and found that they have an irreplaceable role in improving the content of our articles. We cited them in this paper:72、73、74.

2.Thank you very much for the following publications. We have carefully studied these articles and found that they have an irreplaceable role in improving the content of our articles. We cited them in this paper:72、73、74.

3.The purpose of this study based on self-collected data set is to verify the ability of the MI-BCI system to decode 6-classification MI-EEG signals online, and to solve the problems of individual differences and poor performance of cross-subject classification models. The self-collected data set used in this paper has been uploaded: https://physionet.org/content/eegmmidb/1.0.0/

According to the suggestion to use public dataset, this paper has added 4.4 Comparison with other works. The public PhysioNet 109-subject dataset was selected for experiments in this section, and a total of 10 sets of experiments were conducted. The average value was taken as the classification result. The average classification accuracy reached 97.83%, which is better than the research results of other literature listed in Table 9. The results indicate that the method proposed in this paper is effective in motion intention classification and can improve the decoding ability of MI-EEG signals.

4.The left fist, right fist, left foot, right foot, left thumb, and right thumb mentioned in the article are the body movements imagined by the subject, such as "left fist clenched", "right fist clenched", "extend left foot", "extend right foot", "erect left thumb", and "erect right thumb". For example, during training, in order to complete the "left fist" task, a picture of the left fist appears on the screen, and the subject needs to imagine the body movements of the left fist clenching according to the picture.

5.Based on the 6-classification self-collected MI dataset, this paper chooses a smart car as the external controlled device, and builds a MI-BCI system, which can decode the EEG signals of the subjects online and convert them into the corresponding control instructions of the car, thus realizing the brain control function of the car. In other words, the operation result of the intelligent car is a visual way of the classification result of the EEG signal, which lays the foundation for the performance optimization and application of the BCI system in the next step. The corresponding control instructions of the 6 MI tasks of the BCI system are shown in Table 3, and the experimental results of the BCI system are shown in Section 4.3.

6.Thank you for pointing this out. This article regards denoising as a regular part of MI-EEG analysis. We have adjusted the structure of the article according to the suggestion, adding 3.1 denoising results, and retaining a detailed analysis of the denoising effect in section 4.1.

7.Thank you for the valuable suggestion. According to the suggestion, this paper has added 4.4 Comparison with other works. The public PhysioNet 109-subject dataset was selected for experiments, and a total of 10 sets of experiments were conducted. The average value was taken as the classification result. The classification accuracy of the method proposed was compared with other literature research results, as shown in Table 9 in the revised manuscript.

Handiru V S et al. proposed an Iterative Multiobjective Optimization for Channel Selection (IMOCS) algorithm. On the PhysioNet dataset, the SVM classifier achieved an average classification accuracy of 63.62% for the 2 MI task. Youngjoo K et al. proposed a the strong uncorrelating transform complex common spatial patterns (SUTCCSP) algorithm. The performance of multiple classifiers was evaluated based on PhysioNet's 2 MI dataset, in which the random forest (RF) classifier achieved the best classification accuracy of 80.05%. Ma X et al. proposed a method based on Recurrent Neural Networks (RNNs) that can perform parallel decoding of spatial and temporal information. An average classification accuracy of 68.20% was achieved on PhysioNet's 4 MI dataset. An end-to-end DL model constructed by Hauke Dose et al. The CNN model classifies the original MI-EEG signals without special feature extraction. On PhysioNet's 2, 3, and 4 MI datasets, the mean accuracy reached 86.49%, 79.25%, and 68.51%, respectively. Hou Y et al. used CNN model for classification based on Brain source Imaging (ESI), and achieved an accuracy of 94.5% on the 4 MI dataset of PhysioNet. A hybrid optimization technique of Flower Pollination algorithm and β-hill-climbing algorithm proposed by Alyasseri Z et al. An accuracy of 96.05% was obtained using SVM classifier.

In this paper, a combined virtual electrode-based ESA and CNN method is proposed, with an average classification accuracy of 97.83%, which is better than the classification performance of other literatures in the Table 9, indicating that this method is effective in motion intention classification and can improve the decoding ability of MI-EEG signals.

For additional minor improvements:

Our revised version has removed the detailed elaboration of some basic concepts and we have added their DOI identifiers to the references in the revised manuscript.

We have simplified some complex sentences and proofread the whole text.

Our final manuscript will undergo a pay editing service according to the needs of journal: https://www.mdpi.com/authors/english.

Reviewer 3 Report

Comments and Suggestions for Authors

Would be great if you are able to make the data and code available online.

Author Response

Thank you for the valuable suggestion.

The self-collected data set used in this paper has been uploaded: https://physionet.org/content/eegmmidb/1.0.0/

For a revised manuscript, please see the attachment.

Reviewer 4 Report

Comments and Suggestions for Authors

1. The authors assert that the proposed method is effective. However, to what extent has the accuracy improved in comparison to related studies?

For example, we recommend comparing the following reviews.

Altaheri, H., Muhammad, G., Alsulaiman, M., Amin, S. U., Altuwaijri, G. A., Abdul, W., ... & Faisal, M. (2023). Deep learning techniques for classification of electroencephalogram (EEG) Motor imagery (MI) signals: A review. Neural Computing and Applications, 35(20), 14681-14722.

Author Response

Thank you for the valuable suggestion. According to the suggestion, this paper has added 4.4 Comparison with other works. The public PhysioNet 109-subject dataset was selected for experiments, and a total of 10 sets of experiments were conducted. The average value was taken as the classification result. The classification accuracy of the method proposed was compared with other literature research results, as shown in Table 9 in the revised manuscript.

Handiru V S et al. proposed an Iterative Multiobjective Optimization for Channel Selection (IMOCS) algorithm. On the PhysioNet dataset, the SVM classifier achieved an average classification accuracy of 63.62% for the 2 MI task. Youngjoo K et al. proposed a the strong uncorrelating transform complex common spatial patterns (SUTCCSP) algorithm. The performance of multiple classifiers was evaluated based on PhysioNet's 2 MI dataset, in which the random forest (RF) classifier achieved the best classification accuracy of 80.05%. Ma X et al. proposed a method based on Recurrent Neural Networks (RNNs) that can perform parallel decoding of spatial and temporal information. An average classification accuracy of 68.20% was achieved on PhysioNet's 4 MI dataset. An end-to-end DL model constructed by Hauke Dose et al. The CNN model classifies the original MI-EEG signals without special feature extraction. On PhysioNet's 2, 3, and 4 MI datasets, the mean accuracy reached 86.49%, 79.25%, and 68.51%, respectively. Hou Y et al. used CNN model for classification based on Brain source Imaging (ESI), and achieved an accuracy of 94.5% on the 4 MI dataset of PhysioNet. A hybrid optimization technique of Flower Pollination algorithm and β-hill-climbing algorithm proposed by Alyasseri Z et al. An accuracy of 96.05% was obtained using SVM classifier.

In this paper, a combined virtual electrode-based ESA and CNN method is proposed, with an average classification accuracy of 97.83%, which is better than the classification performance of other literatures in the Table 9, indicating that this method is effective in motion intention classification and can improve the decoding ability of MI-EEG signals.

For a revised manuscript, please see the attachment.

Round 2

Reviewer 2 Report

Comments and Suggestions for Authors

The paper still needs to undergo English language editing, but otherwise the others have sufficiently improved the paper.

Comments on the Quality of English Language

The paper still needs to undergo English language editing.

Author Response

Thank you for your sincere suggestions. We have simplified some complex sentences and proofread the whole text.

Our final manuscript will undergo a pay editing service according to the needs of journal: https://www.mdpi.com/authors/english.

For a revised manuscript,please see the attachment.
